# Prognostic Genomic Tissue-Based Biomarkers in the Treatment of Localized Prostate Cancer

**DOI:** 10.3390/jpm12010065

**Published:** 2022-01-07

**Authors:** Gianluca Ingrosso, Emanuele Alì, Simona Marani, Simonetta Saldi, Rita Bellavita, Cynthia Aristei

**Affiliations:** 1Radiation Oncology Section, Department of Medicine and Surgery, Perugia General Hospital, University of Perugia, 06156 Perugia, Italy; emanuele.ali.06@gmail.com (E.A.); maranisimona@libero.it (S.M.); cynthia.aristei@unipg.it (C.A.); 2Radiation Oncology Section, Perugia General Hospital, 06156 Perugia, Italy; saldisimonetta@gmail.com (S.S.); rita.bellavita@ospedale.perugia.it (R.B.)

**Keywords:** localized prostate cancer, prognostic factors, tissue-based biomarkers

## Abstract

**Simple Summary:**

In clinically localized prostate cancer, risk stratification (low-, intermediate- and high-risk) is crucial for the management of such a heterogenic disease, and it is based only on clinicopathologic features (i.e., baseline prostate-specific antigen (PSA), Gleason score and clinical stage of the tumor). New prognostic tools have been developed, mainly based on genomic tissue analysis. The aim of the present overview report is to focus on commercially available tissue-based biomarkers and more specifically on mRNA-based gene expression classifiers: Decipher (GenomeDX Biosciences), Prolaris (Myriad Genetics), and Oncotype Dx (Genomic Health). These new prognostic tests are going to be incorporated in clinicopathologic nomograms to better design the individualized treatment strategy for the cure of localized prostate cancer.

**Abstract:**

In localized prostate cancer clinicopathologic variables have been used to develop prognostic nomograms quantifying the probability of locally advanced disease, of pelvic lymph node and distant metastasis at diagnosis or the probability of recurrence after radical treatment of the primary tumor. These tools although essential in daily clinical practice for the management of such a heterogeneous disease, which can be cured with a wide spectrum of treatment strategies (i.e., active surveillance, RP and radiation therapy), do not allow the precise distinction of an indolent instead of an aggressive disease. In recent years, several prognostic biomarkers have been tested, combined with the currently available clinicopathologic prognostic tools, in order to improve the decision-making process. In the following article, we reviewed the literature of the last 10 years and gave an overview report on commercially available tissue-based biomarkers and more specifically on mRNA-based gene expression classifiers. To date, these genomic tests have been widely investigated, demonstrating rigorous quality criteria including reproducibility, linearity, analytical accuracy, precision, and a positive impact in the clinical decision-making process. Albeit data published in literature, the systematic use of these tests in prostate cancer is currently not recommended due to insufficient evidence.

## 1. Introduction

In clinically localized prostate cancer (PCa), risk stratification (low-, intermediate- and high-risk) is based on baseline prostate-specific antigen (PSA), Gleason score and clinical stage of the tumor [1]. These clinicopathologic variables have been used to develop nomograms (e.g., Partin tables, Briganti nomogram) quantifying the probability of locally advanced disease (i.e., extracapsular extension, seminal vesicles involvement), and of pelvic lymph node and distant metastasis at diagnosis of localized PCa [2,3,4]. Some other calculators such as the Cancer of the Prostate Risk Assessment (CAPRA) or the Stephenson nomogram are used in the post-operative setting to predict the probability of recurrence after radical prostatectomy [5,6]. In the localized setting, these risk assessment tools are of pivotal importance for the management of PCa patients who can be cured with a wide spectrum of treatment strategies including active surveillance (AS), radical prostatectomy (RP) and radiation therapy (RT) [1,7]. Although essential in the daily clinical practice, nomograms based on clinical parameters do not allow the precise distinction of an indolent instead of an aggressive disease [8]. Prognostic biomarkers estimating the likelihood of an adverse outcome, combined with the currently available prognostic tools, might help in the decision-making process providing more tailored treatment for an individual patient. In recent years, several urine, blood, and tissue-based biomarkers have been introduced. In the following overview report, we focus on commercially available tissue-based biomarkers and more specifically on mRNA-based gene expression classifiers: Decipher (GenomeDX Biosciences, San Francisco, CA, USA), Prolaris (Myriad Genetics, Salt Lake City, UT, USA), and Oncotype Dx (Genomic Health, Redwood City, CA, USA).

## 2. Materials and Methods

We reviewed the current literature and gave an overview report on commercially available genomic tissue-based biomarkers in patients affected by localized PCa. We limited the scope of our search to Decipher, Prolaris, and Oncotype Dx because of the current availability of data in the literature about these genomic tests demonstrating rigorous quality criteria, including reproducibility, linearity, analytical accuracy, and precision [9,10]. We performed a PubMed literature search according to the preferred reporting items and meta-analysis (PRISMA) guidelines [11] of the available data for each selected biomarker. Keywords used were: “prostate cancer” or “prostatic cancer” or “prostatic carcinoma” or “prostate carcinoma” and “tissue-based biomarker” or “genetic tissue-based biomarker” or “genomic tissue-based biomarker” or “tissue-based markers”. Our inclusion criteria were as follows: full articles in the English language published within the last 10 years up to 31 May 2021. Titles and abstracts were used to screen for initial study inclusion. Clinical studies published in English language journals were identified and screened for duplicates. Reference lists of the retrieved reports were also manually searched and cross-referenced to ensure completeness. Once a comprehensive list of abstracts has been retrieved and reviewed, any studies meeting inclusion criteria were obtained and reviewed in full. Reviews, commentaries, letters, and conference abstracts were excluded. Two authors (E.A. and S.M.) independently performed the study selection. Disagreements were resolved by consensus with two authors (G.I. and C.A.). We reviewed the full version of each article. The flowchart of the systematic review is reported in [Fig jpm-12-00065-g001]. Data extraction was completed independently by two reviewers (E.A. and S.M.) to establish inter-rater reliability using a standardized form to obtain: (1) general information, author name, year and type of publication, literature source; (2) clinical data, including number of patients, patients’ subset, analyzed tissue-type, and follow-up; (3) study endpoints and statistical methods. Disagreements were resolved by discussion and re-review of the literature. Data were summarized in evidence tables and described in the text.

For risk of bias assessment, we used the star-based Newcastle Ottawa Scale (NOS) (Appendix A). A maximum of one star can be given for each item, except for comparability, for which one or two stars can be given. The risk of bias was considered as low, intermediate, or high for the scores ≥ 7–9, 4–6, and <4, respectively (Appendix A).

Data about the predictive power of each single tissue-based biomarker were extracted and reported in tables. These included concordance index (c-index), which is a measure of goodness of fit for binary outcomes in a logistic regression model, corresponding to the area under the receiver operating characteristic (ROC) curve (ranges from 0.5 to 1). Hazard ratios (HRs) were also used in several studies to define the prediction of biochemical recurrence or prostate cancer specific survival. Eventually, we performed the weighted average of c-indices and of HRs to summarize results.

## 3. Data Synthesis

### 3.1. Decipher

Decipher (Decipher Biosciences, San Diego, CA, USA) is a genomic classifier (GC) of a 22-gene panel predicting the probability of metastatic progression after primary treatment for localized PCa (Table 1). It is a tissue-based assay obtained from formalin-fixed paraffin embedded (FFPE) primary prostate cancer [12], which was developed using a high-density transcriptome-wide microarray analysis.

More specifically, PCa cancer tissue specimen from 545 patients undergone RP at Mayo Clinic between 1987 and 2001 were analyzed profiling the expression of about 1.4 million RNA features. After training and validation sets, selected features were assembled into a classifier using a random forest algorithm. The final GC was based on the expression of 22 RNA biomarkers involved in cell proliferation and differentiation, cell cycle progression, androgen receptor signaling, cell structure and adhesion, immune response (LASP1, IQGAP3, NFIB, S1PR4, THBS2, ANO7, PCDH7, MYBPC1, EPPK1, TSBP, PBX1, NUSAP1, ZWILCH, UBE2C, CAMK2N1, RABGAP1, PCAT-32, GLYATL1P4, PCAT-80, TNFRSF19). The analysis performed after a median follow-up of 16.9 years demonstrated that the GC could better predict metastasis onset than clinical variables alone, and that it could be used as a prognostic tool assessing the probability of systemic progression after primary treatment with a score range between 0 and 1, patients with a score > 0.6 having a high-risk of developing metastatic disease.

In the post-RP setting, Decipher has been tested by several researchers to predict the development of distant metastases (Table 2) compared with clinical variables and with clinical-derived nomograms such as the Stephenson and the CAPRA. Many of the studies are retrospective and monocentric focusing on the subset of high-risk disease, like the one by Karnes et al. [13] evaluating the efficacy of Decipher in the 5-year metastasis prediction after RP in 219 high-risk patients, compared with clinical variables. Using survival receiver operating characteristic (ROC) curves assessing classifier discrimination, the c-index for Decipher was 0.79 (95%CI, 0.68–0.87) outperforming clinical variables.

In the study by Karnes [13] (Table 2) as well as in the others reported in the literature [12,13,14,15,16,17,18,19], Decipher was the predominant predictor of metastasis at multivariate analysis. In the series by Thomas Jefferson University [14], the GC has been retrospectively tested in 139 patients affected by adverse risk factors (pT3 stage or positive margins) after RP and treated with post-operative radiotherapy. The c-index for distant metastasis endpoint was 0.78 (95%CI, 0.64–0.91) for Decipher compared with 0.70 (95%CI, 0.49–0.90) for the post-RP Stephenson model and 0.65 (95%CI, 0.44–0.86) for CAPRA-S. Stratifying patients by the three GC risk-groups (low-risk: <0.4; intermediate-risk: 0.4–0.6; high-risk: >0.6), the 8-years cumulative incidence of distant metastases was 0%, 12% and 17%, respectively (*p* = 0.032). Eventually, high-risk (score > 0.6) patients with undetectable PSA ( ≤ 0.2 ng/mL) before post-operative RT had a distant metastasis cumulative incidence of 3% compared with a rate of 23% for those with detectable PSA (*p* = 0.03).

Taking into account all the c-indices of retrospective studies on the post-operative setting, the weighted average of c-indices was 0.77.

### 3.2. Decipher Role in Clinical Practice

After RP, the use of effective prognostic tools in clinical practice might have an important impact on decision-making for therapy intensification based on the estimated risk of disease recurrence. For instance, Gore et al. [20] prospectively evaluated the effect of Decipher on treatment recommendation in the adjuvant (ART) and salvage (SRT) settings, revealing that high Decipher score was associated with treatment intensification. Eventually, the GC score was an independent predictor for change in management for ART and SRT, at multivariate analysis [20].

Decipher has also been tested in localized PCa to improve prognostication for primary treatment decision-making.

Recently, the applicability of this genomic test in biopsy-derived tissue has been demonstrated, with a high correlation between information derived from RP and biopsy specimens [21]. Klein et al. [17] evaluated at eight years of follow-up the ability of the GC in predicting metastasis from needle biopsy-derived tumor tissue of 57 patients affected by localized PCa (Table 2). The combination of Decipher and National Comprehensive Cancer Network (NCCN) predictive models had an improved c-index of 0.88 (95%CI, 0.77–0.96) compared to NCCN alone (C-index 0.75, 95%CI 0.64–0.87). On multivariate analysis, the GC was the only significant predictor of metastasis when adjusting for age, preoperative PSA and biopsy Gleason score [17].

In 2018, Spratt et al. [22] proposed a three-tier clinical-genomic risk grouping system of distant metastasis and PCa-specific mortality (PCSM) based on genomic and clinicopathological features, demonstrating that Decipher consistently improves prognostic performance over clinicopathological (NCCN classification, and CAPRA-score) variables alone. On a total cohort of 6928 patients studied for development and validation of the prognostic scoring system, c-indices for the three-tier (low-, intermediate-, and high-risk) clinical-genomic risk grouping system significantly outperformed those of NCCN and CAPRA, with 30% of patients being reclassified.

Testing Decipher on biopsy cores, a multi-institutional study on 855 patients affected by localized PCa showed that a high-risk GC score was independently associated with shorter time to treatment in those undergone AS, and with a worse time to failure in those undergone radical therapy [22].

### 3.3. Prolaris

The Prolaris (Myriad Genetics, Salt Lake City, UT, USA) is a multigene test commercially available in the USA and in Europe which uses prostate tissue samples from biopsy or prostatectomy to give prognostic information about PCa (Table 1). It measures the expression of 31 cell cycle progression (CCP) genes with a score range from 0 to 10, a high score correlating with tumor aggressiveness and with the risk of progression. Cuzick et al. first tried to build a CCP score by a gene signature in order to improve PCa patients’ stratification risk [23]. The rationale for the development of such a CCP score is based on the assumption that the measurement of actively growing cells (showing high CCP score) within a tumor gives information about disease aggressiveness and prognosis [10,23].

In the study by Cuzick et al. [23], 126 CCP genes from the Gene Expression Omnibus database were tested on 96 commercially available FFPE PCa sections, creating a gene signature with 31 selected cell cycle genes (FOXM1, CDC20, CDKN3, CDC2, KIF11, KIAA0101, NUSAP1, CENPF, ASPM, BUB1B, RRM2, DLGAP5, BIRC5, KIF20A, PLK1, TOP2A, TK1, PBK, ASF1B, C18orf24, RAD54L, PTTG1, CDCA3, MCM10, PRC1, DTL, CEP55, RAD51, CENPM, CDCA8, and ORC6L), and a predefined score of disease outcome prediction. The genetic signature was then assessed retrospectively in two localized PCa patients’ cohorts (366 patients undergone RP, and 337 patients with PCa diagnosis performed by transurethral resection of the prostate (TURP) undergone watchful waiting).

In the post-prostatectomy setting, the increase in hazard ratio (HR) for a 1-unit change in CCP score proved to be predictive of biochemical recurrence in both univariate (HR 1.89 95%CI 1.54–2.31, *p* = 5.6 × 10^−9^) and multivariate analysis (HR 1.77 95%CI 1.40–2.22, *p* = 4.3 × 10^−6^). Similarly, in the TURP setting the CCP score was strictly related to cancer specific survival (HR 2.57 95%CI 1.93 to 3.43, *p* = 8.2 × 10^−11^) [23]. Up to date, several research works have focused on the use of CCP as a prognostic tool for PCa management and it has been tested on tissue samples deriving not only from RP or TURP but also from biopsies. Bishoff et al. [20,21,22,23,24] retrospectively tested the CCP score on biopsy specimens from three cohorts (283 patients from Martini Clinic, 176 from Durham Veterans Affairs Medical and 123 from Intermountain Healthcare), with a total of 582 localized PCa patients treated with RP (Table 3). For each cohort, at multivariate analysis the CCP score proved to be a strong predictor of biochemical recurrence and metastatic disease (Table 3). One of the limitations of this study was the use, in one of the three cohorts, of simulated biopsies resulting from post-operative tissue blocks. However, the combined analysis carried out excluding this cohort confirmed the CCP score as a strong predictor of biochemical recurrence (BCR), both at univariate (HR 1.45, 95%CI 1.18–1.79, *p* = 5.7 × 10^−4^) and multivariate analyses (HR 1.40, 95%CI 1.1–1.74, *p* = 0.0032). A strong association was also found for metastatic disease but only on univariate analysis (HR 4.69, 95%CI 2.28–9.64, *p* = 1.6 × 10^−5^) [24].

Similar results in terms of BCR prediction were reported by Freedland et al. evaluating the Prolaris test on biopsy samples from 141 localized PCa patients treated with external beam radiotherapy as primary curative therapy (Table 3). The authors obtained a strong correlation between high CCP score and biochemical recurrence (HR for BCR of 2.55 for 1-unit increase in CCP score), which was confirmed at multivariate analysis after adjustments for pretreatment PSA level, Gleason, percent positive cores, and concurrent androgen deprivation therapy (Table 3) [25].

As the stratification risk of localized PCa patients is mainly based on clinical parameters such as preoperative PSA, pathologic Gleason score and pathologic parameters, several authors tried to find an association between the Prolaris test and clinical nomograms such as CAPRA score in order to improve the therapeutic decision-making process. Cooperberg et al. showed in 413 patients undergone RP the usefulness of CCP score to stratify patients with low clinical risk defined by CAPRA score ≤ 2 (HR 2.3, 95%CI 1.4–3.7), moreover they validated a combined CAPRA + CCP score that proved to be more predictive than the CAPRA score alone (*p* < 0.001) [26]. Another validation study conducted on biopsy samples from 585 patients affected by localized PCa reported at multivariate analysis adjusted for clinical parameters a strong correlation of CCP score with cancer-specific survival evaluated as primary endpoint (HR 2.17, 95%CI 1.83–2.57, *p* < 0.0001) (Table 3). The authors also validated the clinical-cell-cycle-risk (CCR) score, which resulted from a combination of CAPRA and CCP score showing a strict relation with death from prostate cancer (HR for a 1-unit change in CCR score 2.17, 95%CI 1.83–2.57; *p* = 4.1 × 10^−21^) [27]. Recently, Canter et al. evaluated CCP and CCR scores as predictive factors for clinical outcomes after prostate cancer treatment. They analyzed 4 cohorts with 1062 patients undergone RP, using biopsy or simulated biopsy samples for Prolaris testing. The authors showed that CCP and CCR score were strictly related to progression to metastatic disease at univariate and at multivariate analysis adjusted for all significant variables (HR for a 1-unit change in CCP score 2.21, 95%CI 1.64–2.98, *p* = 1.9 × 10^−6^; HR for a 1-unit change in CCR score 3.63, 95%CI 2.60–5.05, *p* = 2.1 × 10^−16^) [28]. Regarding BCR, the weighted average of HRs of available studies was 1.68.

### 3.4. Prolaris Role in Clinical Practice

Many studies support the hypothesis that Prolaris test could be used to stratify PCa patients and to guide the therapeutic approach. Several outcomes have been tested, such as BCR and distant metastasis survival (DMS) and a strict association with CCP score has been reported. Prolaris was evaluated on different specimen types (biopsy or RP samples) with no reported differences in terms of predictive utility of the test, although it seems of pivotal importance to adequately perform prostate biopsy in order to reduce the risk of under-sampling errors, which might affect the GC validity [29,30].

In localized PCa, Prolaris might be used as a treatment decision-making tool for primary therapy. For instance, it might help in the better definition of low-risk patients otherwise defined as intermediate or high-risk according to purely clinical variables such as Gleason score, PSA, Ki67 or CAPRA score and this is crucial for the therapeutic decision-making process because an active treatment could be avoided, limiting adverse events and improving patients’ quality of life without neglecting the curative intent.

### 3.5. Oncotype

Oncotype DX (Genomic Health, Redwood City, CA, USA) platform is made up of multi-gene real-time polymerase chain reaction (RT-PCR) assays (Oncotype DX^®^ Assays) used in the treatment-decision process for patients affected by breast or colon cancer. It was first evaluated on a retrospective series of hormone-responsive breast cancer patients with negative lymph nodes, randomly assigned to placebo vs. tamoxifen or tamoxifen vs. cyclophosphamide, methotrexate, fluorouracil, and tamoxifen (CMFT) [31,32]. The risk (i.e., low-, intermediate- and high-risk) of relapse in these 2 cohorts was evaluated based on the expression level of 21 genes on reverse transcriptase-polymerase chain reaction (RT-PCR) from tissue blocks of the primary tumor [32].

In prostate cancer, Oncotype DX (Table 1) integrates with traditional clinical and pathological diagnostic features (PSA, Gleason score, cTNM) in order to better discriminate between indolent and aggressive disease. Compared with the Oncotype DX platform used for breast cancer, the multiplex preamp step has been introduced to create more copies of the initial RNA prior to quantitative evaluation of gene expression allowing the processing of very small samples (5 µm sections). The test analyzes the expression of 17 genes (five housekeeping genes and 12 genes related to prostate cancer) through RT-PCR on fixed, FFPE prostate needle biopsy tissue. The five housekeeping genes (ARF1, ATP5E, CLTC, GPS1, and PGK1) were selected for their low inter-patient variability, lack of relationship to clinical outcome, and robust analytical performance. The 12 cancer-related genes belong to four distinct biological pathways with a role in prostate tumorigenesis: the androgen pathway (AZGP1, KLK2, SRD5A2, and FAM13C), cell organization (FLNC, GSN, TPM2, and GSTM2), proliferation (TPX2) and stromal response (BGN, COL1A1 and SFRP4) (Table 1). The combination of the expression of these genes is used to calculate the Genomic Prostate Score (GPS), which ranges from 0 to 100.

Knezevic et al. demonstrated that Oncotype DX is able to reliably and accurately measure gene expression over a wide range of PCa populations and using very small amounts of RNA, based on the average amplification efficiency of the 17 gene tests (93%), a high analytical sensitivity, and a wide linear range and low bias (less than 9.7%) [33].

In localized PCa, Oncotype DX has been clinically validated to predict the risk of disease recurrence and of prostate cancer death [34]. In the clinical validation study, Oncotype DX was tested using three cohorts of patients: a prostatectomy (*n* = 441), a biopsy (*n* = 167), and a prospectively designed, independent clinical validation cohort (*n* = 395). GPS predicted high-grade (odds ratio [OR] per 20 GPS units: 2.3; 95% confidence interval [CI], 1.5–3.7; *p* < 0.001) and high-stage (OR per 20 GPS units: 1.9; 95%CI, 1.3–3.0; *p* = 0.003) at RP pathology [34]. Cullen et al. demonstrated that Oncotype DX predicts time to biochemical recurrence at univariate analysis (hazard ratio per 20 GPS units [HR/20 units]: 2.9; *p* < 0.001) and time to metastases (HR/20 units: 3.8; *p* = 0.032) after primary treatment [35]. In a RP retrospective cohort of 279 localized PCa patients, Van Den Eeden et al. assessed the association between Oncotype DX and time to metastases and PCSM (Table 4). Analyzing a total of 259 GPS valid results, they demonstrated a strong correlation of GPS score with the two endpoints. Moreover, at the time of analysis (median follow-up 9.8 years) no patient with low- or intermediate risk and GPS score < 20 developed metastases or died of PCa. At ROC analysis, the combination of Oncotype DX with CAPRA score significantly improved the c-statistic of CAPRA alone from 0.65 to 0.73 for metastasis prediction and from 0.78 to 0.84 for CSM [36].

In a retrospective analysis testing GPS on 428 patients undergone RP between 1987 and 2004, GPS score was significantly associated with the risk of distant metastasis and CSM at 20 years of follow-up. Eventually, GPS score < 20 indicated a low risk of both outcomes, whereas a score > 40 indicated a high-risk of developing distant metastases and of dying of PCa [37]. For PCSM, the weighted average of AUC at ROC curves was 0.81 and for adverse pathology after RP was 0.76.

### 3.6. Oncotype DX Role in Clinical Practice

Oncotype DX has been used to better discriminate between indolent and aggressive disease in the primary setting as well as in the post-operative setting. For instance, Moschovas et al. investigated the capability of Oncotype DX in predicting adverse pathological features (ie extraprostatic extension (EPE), positive surgical margin (PSM) and seminal vesicle invasion (SVI)) in patients treated with RP for localized PCa (Table 4). Multivariate analysis assessing the odds ratio per 20-points change in Oncotype DX genomic score showed that GPS is an independent predictor of adverse pathological features after RP, and specifically for EPE and SVI. At ROC analysis, GPS score did not increase the area under the curve (AUC) of PCSM [38].

Recently, Oncotype DX has been tested as a predictor of outcome for patients in active surveillance. In the PASS trial, GPS scores available in 432 patients were evaluated for the association with adverse pathological features at RP after a period of active surveillance. At the time of analysis, on total of 101 RP with central pathology review Oncotype DX was not significantly associated with adverse pathological features neither with upgrading in surveillance biopsy [39].

## 4. Conclusions

A prognostic biomarker must estimate the likelihood of a disease characteristic being present or absent more accurately determining the prognosis. Molecular information, providing specific insights into the underlying tumor biology, combined with clinicopathologic features might improve the decision-making process and clinical outcomes. In recent years, tissue-based mRNA-GC have been widely investigated as new tools for localized PCa prognosis. More specifically, they have been tested in the context of newly diagnosed prostate cancer and of surgically treated patients, in order to better define risk stratification and to guide clinical management especially in borderline scenarios such as AS for specific subsets of localized PCa patients or treatment intensification after RP.

In our systematic review, Decipher, Prolaris, and Oncotype Dx, which are commercially available tissue-based biomarkers demonstrating rigorous quality criteria, seem to be reliable prognostic tools for the prediction of biochemical recurrence or prostate cancer specific survival. Despite advances in tissue-based mRNA-GC validation and data published in literature, the systematic use of these tests in prostate cancer is currently not recommended due to insufficient evidence. About validation, many of the results are based on White Caucasian cohorts. About data published in the literature, evidence of efficacy derives from retrospective monocentric studies with short median follow-up and low number of events. Prospective randomized trials are needed for the safe and effective use of these tools in clinical practice [10]. Moreover, tissue-based biomarkers results have an important intrinsic limitation coming from their dependence on the sampled tumor, showing PCa multifocality and high intratumoral heterogeneity [30]. To date, no comparison studies between tissue-based GCs have been published and there is currently uncertainty regarding the potential specific role of each available biomarker. The American Society of Clinical Oncology (ASCO) guidelines on molecular biomarkers in localized PCa, which have been recently published, recommend the use of tissue-based biomarkers “only in situations in which a specific assay result, when considered in combination with routine clinical factors, will clearly affect the management decision” [40].

In the future, it is likely these prognostic biomarkers will be incorporated in clinicopathologic nomograms to better design the personalized diagnostic and treatment strategy for each single patient.

## Figures and Tables

**Figure 1 jpm-12-00065-g001:**
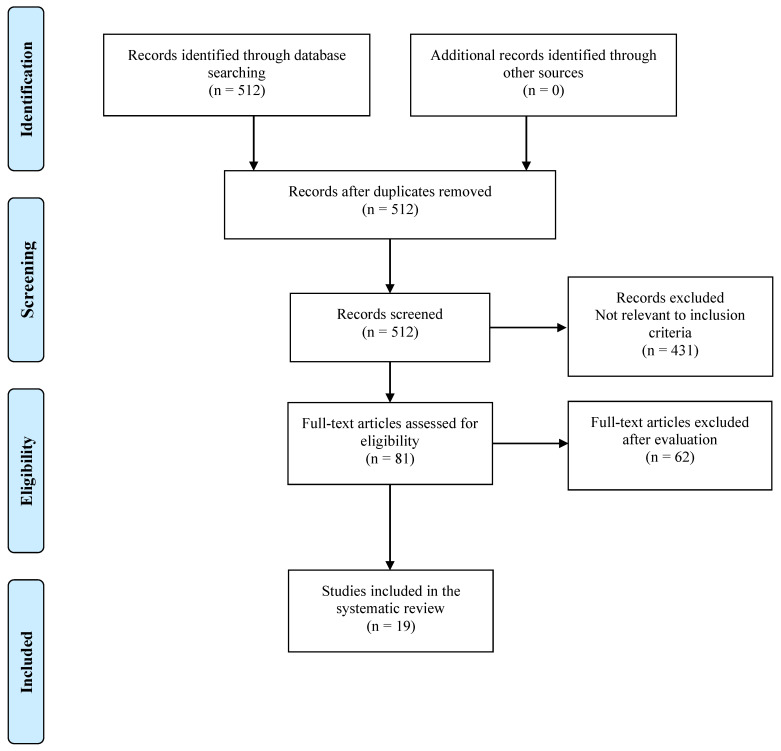
Flowchart of inclusion of studies in the systematic review.

**Table 1 jpm-12-00065-t001:** Tissue-based mRNA genomic classifiers.

Tissue Biomarker	Laboratory	Tested Genes	Score Report	Clinical Use
Decipher 22 genes	GenomeDx (San Diego, CA, USA)	LASP1, IQGAP3, NFIB, S1PR4, THBS2, ANO7, PCDH7, MYBPC1, EPPK1, TSBP, PBX1, NUSAP1, ZWILCH, UBE2C, CAMK2N1, RABGAP1, PCAT-32, GLYATL1P4, PCAT-80, TNFRSF19	GC score: 0–1	Post-RP: to predict the probability of disease recurrence after primary treatment. At localized PCa diagnosis: to categorize patients into risk groups and better define for AS vs. treatment and treatment intensification.
Prolaris 31 genes	Myriad Gentics (Salt Lake City, UT, USA)	FOXM1, CDC20, CDKN3, CDC2, KIF11, KIAA0101, NUSAP1, CENPF, ASPM, BUB1B, RRM2, DLGAP5, BIRC5, KIF20A, PLK1, TOP2A, TK1, PBK, ASF1B, C18orf24, RAD54L, PTTG1, CDCA3, MCM10, PRC1, DTL, CEP55, RAD51, CENPM, CDCA8, ORC6L	CCP score: 0–6	To predict the risk of metastasis and CSM. To better define for treatment after primary therapy.
Oncotype Dx 17 genes	Genomic Health, Redwood City, CA, USA	ARF1, ATP5E, CLTC, GPS1, PGK1, AZGP1, KLK2, SRD5A2, FAM13C, FLNC, GSN, TPM2, GSTM2, TPX2, BGN, COL1A1, SFRP4	GPS score: 0–100	To predict the risk of adverse pathological features (EPE, SVI) after RP.

RP: radical prostatectomy; PCa: prostate cancer; AS: active surveillance; CSM: cancer-specific survival; EPE: extra-prostatic extension; SVI: seminal vesicles involvement.

**Table 2 jpm-12-00065-t002:** Decipher studies.

	Study Type	No of Pts	Setting	Tissue Type	Disease State	Median Fu	Endpoint	Decipher c-Index (95%CI)
Erho 2013 [12]	Retrospective, nested-case control (Mayo Clinic)	545	Post-RP	RP	All risk classes	16.9 year	Metastasis prediction	0.75 (0.67–0.83)
Karnes 2013 [13]	Retrospective (Mayo Clinic)	219	Post-RP	RP	High-risk	6.7 year	5-year metastasis prediction compared with clin. variables	0.79 (0.68–0.87)
Cooperberg 2014 [14]	Retrospective (Mayo Clinic)	185	Post-RP	RP	High-risk	6.4 year	PCSM prediction compared with CAPRA-S	0.78 (0.68–0.87)
Ross 2014 [15]	Retrospective (Mayo Clinic)	85	BCR after RP	RP	High risk with BCR	NA	Metastasis prediction compared with clin. variables, CAPRA-S and Stephenson	0.82 (0.77–0.86)
Den 2014 [16]	Retrospective (Thomas Jefferson University)	139	Post-RP + PORT	RP	adverse risk factors after RP	NA	Metastasis and BCR prediction compared with CAPRA-S and Stephenson	0.78 (0.64–0.91)
Klein 2015 [17]	Retrospective (Cleveland Clinic)	169	Post-RP	RP	High-risk	NA	5-year metastasis prediction compared with CAPRA-S and Stephenson	0.77 (0.66–0.87)
Ross 2016 [18]	Retrospective (John Hopkins)	260	Post-RP	RP	Intermediate and high-risk	9 year	Metastasis prediction	0.76 (0.66–0.84)
Den 2015 [19]	Retrospective (Bi-institutional)	188	Post-RP + PORT	RP	adverse risk factors after RP	8 year	Metastasis prediction compared with CAPRA-S	0.83 (0.27–0.89)

RP: radical prostatectomy; PORT: post-operative radiotherapy; PCSM: prostate cancer-specific survival; BCR: biochemical recurrence; NA: not available.

**Table 3 jpm-12-00065-t003:** Prolaris studies.

	Study Type	No of Pts	Setting	Tissue Type	Median Fu	Endpoint	CCP Results
Cuzick 2011 [23]	Retrospective monocentric	366 337	Post-RP Post-TURP	RP TURP	NA	BCR CSS	MVA: HR for a 1-unit change in CCP score 1.77 95%CI 1.40–2.22, *p* = 4.3 × 10^−6^ MVA: HR for a 1-unit change in CCP score 2.56 95%CI 1.85–3.53, *p* = 1.3 × 10^−8^
Bishoff 2014 [24]	Retrospective multicentric	582	Clinically localized	Biopsy	61-88 mo	BCR DMS	MVA: HR 1.47 95%CI 1.23–1.76, *p* < 10^−4^ MVA: HR 4.19 95%CI 2.08–8.45, *p* < 10^−5^
Freedland 2013 [25]	Retrospective monocentric	141	Clinically localized	Biopsy		BCR	MVA: HR for a 1-unit change in CCP score 2.11, 95%CI 1.04–4.25, *p* < 0.034
Cuzick 2015 [27]	Retrospective multicentric	585	Clinically localized	Biopsy	9.52 mo	PCSM	MVA adjusted for CAPRA score: HR for a 1-unit change in CCR score 2.17, 95%CI 1.83–2.57; *p* = 4.1 × 10^−21^
Cooperberg 2013 [26]	Retrospective multicentric	413	Post-RP	RP	85 mo	Biochemical/clinical recurrence	MVA adjusted for CAPRA score: HR for a 1-unit change in CCP score 1.7, 95%CI 1.3–2.3; *p* < 0.001
Canter 2020 [28]	Retrospective multicentric	1062	Post-RP	Biopsy or simulated biopsy		Progression to metastatic disease	MVA adjusted for CAPRA score: HR for a 1-unit change in CCP score 2.21, 95%CI 1.64–2.98; *p* = 1.9 × 10^−6^

RP: radical prostatectomy; CSS: cancer-specific survival; BCR: biochemical recurrence; PCSM: prostate cancer-specific survival; MVA: multivariate analysis; NA: not available.

**Table 4 jpm-12-00065-t004:** Oncotype DX studies.

	Study Type	No of Pts	Setting	Tissue Type	Disease State	Median Fu	Endpoint	Oncotype DX AUC at ROC Curve
Klein 2014 [34]	Retrospective	441	Post-RP	Biopsy	All risk classes	NA	Clinical recurrence, adverse pathology, PCSM	NA
Van Den Eeden 2018 [36]	Retrospective	279	Post-RP	Biopsy	All risk classes	9.8 year	Metastasis and PCSM prediction compared with clinical variables only	Metastasis: 0.73 PCSM: 0.84
Brooks 2021 [37]	Retrospective	428	Post-RP	RP index lesion	All risk classes	15.5 year	Metastasis and PCSM prediction compared with clinical variables only	Metastasis: 0.82 PCSM: 0.82
Covas Moschovas 2021 [38]	Retrospective	749	Post-RP	biopsy	All risk classes	Median time between GPS test and RP: 176 days	Prediction of adverse pathology features (EPE, PSM, SVI) compared with clinical variables only	EPE: 0.70 SVI: 0.78 PCSM: not improved
Cullen 2021 [35]	Retrospective	431	Post-RP	Biopsy	Low-, intermediate-risk	5.2 year	BCR, adverse pathology	Adverse pathology: 0.72 BCR: 0.68

RP: radical prostatectomy; BCR: biochemical recurrence; PCSM: prostate cancer-specific survival; NA: not available.

## Data Availability

Not applicable.

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
