# Peer review of "Prognostic Genomic Tissue-Based Biomarkers in the Treatment of Localized Prostate Cancer"

_jpm, 2022, doi:10.3390/jpm12010065_

Round 1

Reviewer 1 Report

In this manuscript, Ingrosso et al., review the literature of the last 10 years concerning the commercially available genomic tissue-based biomarkers for patients affected by localized prostate cancer. Especially the authors focus their manuscript on the mRNA-based gene expression classifiers.

Overall, this review paper is well written, and the literature offers a useful overview of current research and policy, and the resulting bibliography provides a very useful resource for current practitioners. My following comments are of minor character:

- The first part of the manuscript (before the abstract) is a bit confusing as far as affiliations are concerned. In addition, the following sentences should be deleted: “All authors have made….”, Conflict of interest, funding, Cover letter….

- Introduction: Please use a synonym for “of such heterogeneous disease”, and avoid repetitions such as “radical prostatectomy (PT)”

- Please check and correct the reference list style

- Tables 1-4 should be integrated into the text

Author Response

In this manuscript, Ingrosso et al., review the literature of the last 10 years concerning the commercially available genomic tissue-based biomarkers for patients affected by localized prostate cancer. Especially the authors focus their manuscript on the mRNA-based gene expression classifiers.

Overall, this review paper is well written, and the literature offers a useful overview of current research and policy, and the resulting bibliography provides a very useful resource for current practitioners. My following comments are of minor character:

- The first part of the manuscript (before the abstract) is a bit confusing as far as affiliations are concerned. In addition, the following sentences should be deleted: “All authors have made….”, Conflict of interest, funding, Cover letter….

We amended as requested.

- Introduction: Please use a synonym for “of such heterogeneous disease”, and avoid repetitions such as “radical prostatectomy (PT)”

We amended, as requested.

- Please check and correct the reference list style

We amended, as requested.

- Tables 1-4 should be integrated into the text

We integrated tables in the text.

Reviewer 2 Report

Prostate-Specific Antigen (PSA), Gleason score and clinical stage of the tumor are necessary in clinically localized prostate cancer (PCa) for the risk stratification of the patients. Due to risk stratification, PCa patients could be employed evidence-based strategies including active surveillance (AS), radical prostatectomy (RP) and radiation therapy (RT). Prognostic biomarkers, combined with the currently available prognostic tools, might help in the decision-making process providing a tailored management of PCa patient. In recent years, several urine, blood, and tissue-based biomarkers have been introduced. The aim of the study was to conduct an overview on commercially available tissue-based biomarkers, specifically on mRNA-based gene expression classifiers such as Decipher (GenomeDX Biosciences), Prolaris (Myriad Genetics), and Oncotype Dx (Genomic Health).

  • Authors should be congratulated for the challenging work. All future prospective should lead to improve PCa algorithms to properly manage early stage PCa patients, avoiding overtreatment. Despite the interesting topic, the paper does not add anything new to the current literature. A similar and comprehensive work was already published (PMID: 33339117 DOI: 10.3390/cancers12123790)

Author Response

  • Authors should be congratulated for the challenging work. All future prospective should lead to improve PCa algorithms to properly manage early stage PCa patients, avoiding overtreatment. Despite the interesting topic, the paper does not add anything new to the current literature. A similar and comprehensive work was already published (PMID: 33339117 DOI: 10.3390/cancers12123790)

We thank the Reviewer for his considerations. There are several works on this hot topic. Compared to other published reviews, the present article is specifically focused on tissue-based biomarkers and more specifically on mRNA-based gene expression classifiers.

We hope the Reviewer might reconsider our work.

Reviewer 3 Report

The topic of this study is interesting and important. However, the manuscript can be improved with some addition.

1. I am not sure if this article is review or systematic review/meta-analysis. Because authors mentioned that they followed PRISMA guideline to search literature but they only summarized the related studies. If this is more like to be systematic review, then it is better to add short result part focusing on the screening process and its results. I think basic quantitative analysis including weighted average or comparion on c-index of decipher or HR of CCP result of Prolaris studies, or AUC of oncotype Dx can be possible. 

If this is just a review, it should focus on the mechanismic background or comparison of history and other studies. 

I personally recommend the first one, so I suggest authors to add analysis and statistic analysis in method part, to add PRISMA diagram, and to add result of summarized or combined results of c-index of decipher or HR of CCP result of Prolaris studies, or AUC of oncotype Dx, so that readers can indirectly compare the actual performance and efficacy of these molecular preditive biomarkers on prostatic cancer. It is also necessary to do quality assessment such as NOS scale system on included studies.

2. The discussion and conclusion part can be improved. Generally each paragraph is too long. It is better to devide into shorter paragraphs. please use topic sentence to deliver core messages of each paragraph. It is hard to read and hard to figure out what is the core message and what authors want to say. So, what is your opinion on these tests? Is it trustworthy? then why it is so? The manuscript should give these answer simply and clearly. 

I think it is also important to think about the results of these tests in other cancers briefly. 

Author Response

we thank the reviewer for the comments. We amended the text as requested, adding the PRISMA flow-chart, weighted average of c-indices and HRs. We decided to write a overview report and not a meta-analysis because at the initial meta-analysis data are very heterogeneous and we obtained a I2>70%. We added the NOS table as a supplementary table, as requested.

We divided the paragraphs, as suggested.

We believe that in the "Conclusions" it is clearly stated that "In our systematic review, Decipher, Prolaris, and Oncotype Dx, which are commercially available tissue-based biomarkers demonstrating rigorous quality criteria, seems to be reliable prognostic tools for the prediction of biochemical recurrence or prostate cancer specific survival. Despite advances in tissue-based mRNA-GC validation and data published in literature, the systematic use of these tests in prostate cancer is currently not recommended due to insufficient evidence."

Reviewer 4 Report

The precise distinction of an indolent instead of an aggressive disease in prostate cancer is one of the major goals of prostate cancer research. In this paper, Ingrosso et al. review the recent data on prognostic genomic tissue-based biomarkers in the treatment of localized prostate cancer.

General comment:

The chosen topic is relevant and the article is well-written, concise, and suitable for publication in the Journal of Personalized Medicine.

Minor points:

Instructions for Authors say: ''… Tables should be inserted into the main text close to their first citation''.

The authors should briefly explain what the ''c-index'' and ''c-statistic'' are.

In the following sentence: ''In the study by Karnes (table 2) as well as in the others reported in literature…'' it would be good to put a reference for Karnes, and after ''… in the others reported in literature…'' refer to the ''table 2''. The same for Klein et al.

The abbreviations used in the tables should be defined below the tables. (Instructions for authors say: ''Acronyms/Abbreviations/Initialisms should be defined the first time they appear in each of three sections: the abstract; the main text; the first figure or table.'')

Author Response

we thank the reviewer for the comments.

We amended the text as requested, and added explanation of statistics

Round 2

Reviewer 2 Report

The Paper is not suitable for publication.

Author Response

no notes. We are sorry that reviewer thinks the review is not suitable for publication.